# Correlation of Leptin, Proinflammatory Cytokines and Oxidative Stress with Tumor Size and Disease Stage of Endometrioid (Type I) Endometrial Cancer and Review of the Underlying Mechanisms

**DOI:** 10.3390/cancers14020268

**Published:** 2022-01-06

**Authors:** Clelia Madeddu, Elisabetta Sanna, Giulia Gramignano, Luciana Tanca, Maria Cristina Cherchi, Brunella Mola, Marco Petrillo, Antonio Macciò

**Affiliations:** 1Department of Medical Sciences and Public Health, University of Cagliari, 09042 Monserrato, Italy; clelia.madeddu@unica.it; 2Department of Gynecologic Oncology, ARNAS G. Brotzu, 09100 Cagliari, Italy; elisabetta.sanna@aob.it; 3Medical Oncology Unit, San Gavino Hospital, 09037 San Gavino, Italy; giulia.gramignano@atssardegna.it; 4Medical Oncology Unit, A. Businco Hospital, ARNAS G Brotzu, 09100 Cagliari, Italy; l.tanca@virgilio.it (L.T.); cristina.cherchi@aob.it (M.C.C.); 5Hematology and Transplant Center, A. Businco Hospital, ARNAS G. Brotzu, 09100 Cagliari, Italy; b.mola@studenti.unica.it; 6Gynecologic and Obstetric Unit, Department of Medical, Surgical and Experimental Sciences, University of Sassari, 07100 Sassari, Italy; mpetrillo@uniss.it; 7Department of Surgical Sciences, University of Cagliari, 09042 Monserrato, Italy

**Keywords:** body mass index, adipose tissue, leptin, inflammation, interleukin-6, oxidative stress, endometrioid endometrial cancer, estrogens

## Abstract

**Simple Summary:**

Endometrioid endometrial cancer is typically estrogen-positive and associated with obesity. Indeed, circulating estrogens are strongly and linearly related to adiposity and increased BMI, which have been identified as the most important risk factors for endometrioid endometrial cancer. However, the relationship between excess body weight and endometrial cancer is more complex, and, besides the unopposed estrogens, involves multiple mechanisms, including hyperinsulinemia, altered adipokines, inflammation, and oxidative stress associated with obesity. We investigated the association between classical tumor prognostic factors (i.e., tumor size (T), and nodal (N) and metastatic (M) status) and the levels of leptin, proinflammatory cytokines, and oxidative stress, together with BMI, among type I (endometrioid) and type II endometrial cancer patients. We found that BMI, leptin, IL-6, and reactive oxygen species correlated with T, N, and M status in type I, but not in type II endometrial cancers. This could open new therapeutic perspectives based on the specific pathogenetic mechanisms involved.

**Abstract:**

Endometrioid endometrial cancer is associated with increased BMI and obesity through multiple pathogenetic mechanisms involving hyperestrogenism, hyperinsulinemia, altered adipokine secretion, inflammation, and oxidative stress. In the present study, we aimed to investigate the correlation between BMI, leptin, the proinflammatory cytokines IL-6 and TNFα, reactive oxygen species (ROS), and the traditional prognostic factors T, G, N and M status among type I endometrioid and type II endometrial cancer patients. We enrolled 305 consecutive endometrial cancer patients prospectively. We found that BMI, leptin, and IL-6 significantly correlated with T status, N status, and M status among endometrioid type I endometrial cancer patients. Among type II endometrial cancer patients, BMI and leptin did not correlate with any of the prognostic parameters, whereas there was a positive correlation between IL-6 and the presence of distant metastases. In the multivariate regression analysis, BMI, leptin, and IL-6 were independent predictive variables of T, N, and M status in endometrioid type I endometrial cancer patients. Our study demonstrates that weight gain, adiposity-related adipokines, inflammation, and oxidative stress correlate with the prognostic factors of endometrioid endometrial cancer. Knowledge of the role of obesity-related biological pathways and mediators in the pathogenesis and prognosis of endometrioid endometrial malignancies may offer new perspectives on combined therapeutic strategies that have not been explored to date, both in the advanced disease and in the adjuvant setting.

## 1. Introduction

More than half of endometrial cancers are associated with obesity, which is considered an independent risk factor for this neoplasm [1]. The impact of obesity on different clinical and histological phenotypes is poorly understood, although the association between body mass index (BMI) and endometrial cancer is more prominent for type I cancers, which include typically endometrioid histotypes, than for type II non-endometrioid endometrial cancers [2]. Indeed, unopposed estrogen has been thought to be the primary oncogenic mechanism linking obesity to the development of type I endometrioid endometrial cancer, while type II cancers are usually considered to be estrogen independent [3]. The adipose tissue contains the aromatase enzyme, which peripherally converts circulating androgens, primarily androstenedione, into excess estrogen. This causes continued stimulation of the endometrium, resulting in endometrial hyperplasia, which can subsequently progress to invasive cancer [4]. Accordingly, in endometrioid endometrial cancer patients, the circulating levels of estrogens are strongly and linearly related to adiposity, and weight increase has been identified as the most important risk and prognostic factor for endometrial cancer [5].

However, the relationship between excess body weight and endometrial cancer is more complex and involves multiple biological mechanisms. Obesity associated with metabolic syndrome results in increased levels of insulin and insulin-like growth factor (IGF), which mediate endometrial cancer growth; in turn, hyperinsulinemia is associated with lower levels of sex hormone-binding globulin, which increases the bioavailability of circulating estrogens. Moreover, adipose tissue is well recognized as an active endocrine-immune organ that produces adipokines and proinflammatory mediators, mainly the inflammatory cytokines interleukin (IL)-6 and tumor necrosis factor-alpha (TNF-α), which can promote the development of endometrial cancer [6,7,8,9,10,11]. The likeliest hypothesis is that all the above mechanisms together may contribute to the association between menopause, increased body weight, and hormone-positive endometrial cancer.

Among adipokines, leptin has been studied for its influence on endometrial cancer risk and tumor biology. Leptin, a 16 kD bioactive protein encoded by the Ob gene, exerts its effects through binding to its membrane-bound receptor protein (ObR), and it acts as a regulator of energy and food intake through satiety control in the hypothalamus, as well as modulation of glucose and insulin homeostasis through activation in peripheral tissues [12]. Leptin is released by adipocytes in relation to BMI. Additionally, preadipocytes produce leptin, particularly under paracrine stimulation by proinflammatory cytokines released by adipose tissue macrophages. Considering its multiple endocrine actions, leptin can be viewed as the paradigm of all adipose-derived hormones and adipokines [13].

Chronic inflammation related to obesity can be an important mechanism of endometrial oncogenesis. The plasma concentrations of the proinflammatory cytokines TNF-α and IL-6 positively correlate with BMI, and, in turn, have been directly associated with endometrial cancer promotion and progression [8,14,15]. Indeed, IL-6 has been found to be overexpressed in the stroma of endometrial cancer [16,17]. Thus, inflammation can contribute to the development of endometrial cancer, in conjunction with estrogen exposure.

As a result of inflammatory and metabolic changes, obesity is also associated with oxidative stress. Moreover, hyperglycemia, along with elevated free fatty acid levels, associated with obesity-related metabolic syndrome, induces the production of radical reactive oxygen species (ROS) [18]. It is well known that ROS, in turn, damage lipids, proteins, and nucleic acids, contributing to tumor promotion. Moreover, they are also able to induce the activation of the phosphatidylinositol 3-kinase-Akt-mammalian target of the rapamycin (PI3K/AKT/mTOR) pathway signaling involved in endometrial cancer (EC) proliferation and progression [19,20]. Thus, oxidative stress and the chronic inflammation network are interconnected and together establish a vicious cycle that maintains EC development.

Although the potential role of obesity and adiposity-related hormones, adipokines, inflammation, and oxidative stress in EC development has been studied, currently no therapeutic or preventive approach based on targeting these pathways has been approved and entered clinical practice.

The aim of the present study was to prospectively evaluate the association of established tumor prognostic factors with BMI and blood levels of leptin, proinflammatory cytokines, and oxidative stress markers among patients with type I endometrioid and type II non-endometrioid endometrial cancers. We assessed endometrioid grade 3 cancers together with type II tumors, since several important reports have strongly demonstrated that high-grade endometrioid cancers have risk factors, clinical behavior, and prognosis superimposable to those of non-endometrioid cancers [21,22].

## 2. Materials and Methods

### 2.1. Patients

All consecutive eligible endometrial cancer patients referred to the Department of Gynecologic Oncology, A. Businco Hospital, Cagliari, Italy, were prospectively enrolled in the study between June 2013 and January 2021. Patients were eligible if they had a histologically confirmed diagnosis of endometrial cancer. The exclusion criteria were prior or any other coexisting malignancy, active use of exogenous hormones (hormone replacement therapy), use of medications that could influence diet and lipids, and acute and serious coexisting medical conditions, including serious infections, at the time of blood donation. The research protocol was approved by the Local Institutional Ethics Committee. All subjects participated in the study as volunteers, after signing an informed written consent form. The study was performed under the Declaration of Helsinki.

### 2.2. Measurements

Patients’ body weight, BMI and laboratory parameters were evaluated prior to any surgical or medical antineoplastic treatment. Weight was assessed by a body height–weight balance scale following overnight fasting. BMI was determined as weight (kg)/height (m^2^) and categorized based on World Health Organization standards. Fasting blood samples (10 mL) were collected in the morning, immediately centrifuged, and the serum was stored at −80 °C until the leptin and proinflammatory cytokine assays were performed. Oxidative stress parameters were measured in fresh blood samples. Data on tumor histology, grade (G), size (T), nodal status (N), ER and progesterone (PR) status, and Ki-67 expression were obtained from pathology reports. According to tumor histology, endometrial cancer was classified into type I endometrioid cancer, which is typically associated with hyper-estrogenic states and obesity, and type II non-endometrioid cancer, which includes serous, clear cell, mucinous adenocarcinoma, and mixed cell adenocarcinoma [23,24]. Cancers with other histologies were not included owing to the very low numbers for each type. Endometrioid grade 3 tumors were assessed together with type II non-endometrioid cancer since they behave similarly to type II cancers in terms of prognosis [21,22]. The stage of disease was assessed according to the FIGO 2009 staging system [24].

### 2.3. Assessment of Ki-67 Expression in Tumor Specimen

Whole section Ki-67 expression in tumor specimen was assessed by immunohistochemistry using the Leica Bond Max (Leica Biosystems, Wetzlar, Germany) with heat-induced epitope regaining. Staining was carried out following the protocol indicated by the International Ki-67 in Breast Cancer Working Group [25]. Slides were stained at room temperature for 1 h with the primary antibody (monoclonal mouse, anti-human Ki-67 antibody; DAKO, Carpinteria, CA, USA) at a dilution of 1:100. Then, the primary antibody was detected using the Refine Detection Kit (Leica Biosystems, Wetzlar, Germany), which contains a rabbit anti-mouse IgG secondary antibody and anti-rabbit poly-HRP IgG antibody, and 3,3′-diaminobenzidine was used as a chromogen. The slides were finally counterstained with hematoxylin. The Ki-67 proliferation index was the percentage of positively stained nuclei scored by manual scoring, which calculated the percentage of positively stained nuclei within three high-powered fields (×40 magnification) randomly identified throughout the tumor, assuring no less than 1000 nuclei were counted [26].

### 2.4. Assessment of Serum Leptin, Proinflammatory Cytokines and Oxidative Stress

Serum leptin levels were measured using a double antibody “sandwich” enzyme-linked immunosorbent assay (ELISA) test (DRG Instruments, Marburg, Germany). The absorbance was measured at 450 ± 10 nm. The intra- and inter-assay variations were 5% and 7%, respectively. The results were expressed in nanograms per milliliter (ng/mL). The proinflammatory cytokines IL-6 and TNF-α were detected by a “sandwich” ELISA test (Immunotech SA, Marseille, France). The absorbance at 450 nm was measured with a spectrophotometer (Sirio, Seac, Florence, Italy). Intra-assay variations were 3% for IL-6 and 6% for TNF-α. Inter-assay variations were 7% for TNF-α and 8% for IL-6. The results were expressed in picograms per milliliter (pg/mL). Each assay was performed in duplicate.

Reactive oxygen species were evaluated by the FORT test (Callegari, Parma, Italy), and their value was reported as FORT U. Erythrocyte activity of glutathione peroxidase (GPx) and superoxide dismutase (SOD) was analyzed using a commercial assay (Ransod; Randox Lab, Crumlin, UK), and values were reported as U/L and U/mL for GPx and SOD, respectively. Detailed techniques have already been reported in our previous studies [27,28].

### 2.5. Statistical Analyses

Continuous parameters were checked for linearity and reported as mean and standard deviation. Categorical data were reported as absolute numbers and percentages. The differences between endometrial subtypes were assessed by the Student’s *t*-test for the continuous data and the chi-squared analysis for categorical data. The differences between prognostic categories (pT, G, pN, M status and FIGO stage) were measured through the one-way analysis of variance (ANOVA) or chi-squared tests, as appropriate. The inter-correlations between BMI, leptin, proinflammatory factors (IL-6 and TNF-α), and oxidative stress parameters, as well as between standard tumor prognostic parameters (pT, pN, pG, M status, and stage of disease) and BMI, leptin, proinflammatory cytokines (IL-6 and TNF-α), and oxidative stress markers, were assessed by Pearson’s or Spearman’s rank correlation analysis, as appropriate. Significant associations were assessed by multivariate regression analysis to establish the predictive value of BMI, leptin, proinflammatory cytokines, and oxidative stress markers towards standard prognostic factors. A two-tailed *p*-value was established as significant at a value of <0.05. All statistical analyses were carried out using the SPSS software package, version 21 for Windows (SPSS Inc., Chicago, IL, USA).

## 3. Results

### 3.1. Patients

In total, 305 endometrial cancer patients were enrolled; 83.5% had type I low-grade endometrioid carcinoma, and 16.5% had type II non-endometrioid and G3 endometrioid carcinoma. Table 1 and Table 2 report the patients’ clinical characteristics and the studied tumors’ characteristics, both for the whole group and divided into type I and II cancer subtypes. The proportion of patients with type 2 diabetes at baseline was higher among those with type I disease than for those with type II disease. The patients’ distributions across tumor categories (T, N, and M) did not differ significantly between type I and II endometrial cancer patients. Notably, the G distribution and mean Ki-67 expression differed significantly between type I and II cancers; in fact, the type II group showed an obviously higher incidence of G3 tumors and a higher mean value of Ki-67 expression.

### 3.2. Evaluation of BMI, Leptin, Proinflammatory Cytokines, and Oxidative Stress Parameters According to Histological Subtype

The type I endometrial cancer patients showed a significantly higher BMI and there was a higher proportion of those with obesity than the type II endometrial cancer patients (Table 2). The leptin levels were significantly higher in type I than in type II disease (Table 3). The levels of the proinflammatory cytokine IL-6 were significantly higher in type I disease, whereas those of TNF-α did not differ significantly between type I and II disease (Table 3). The levels of ROS were significantly higher in the type I patients than in the type II endometrial cancer patients. GPx and SOD did not differ significantly between the two patient groups (Table 3).

### 3.3. Evaluation of BMI, Leptin, Proinflammatory Cytokines, and Oxidative Stress Parameters across Tumor Standard Prognostic Categories within Type I and Type II Endometrial Cancer Patients

Among the type I endometrial cancer patients, the BMI and serum leptin levels were significantly higher in those patients with higher values for the T, N, and FIGO stages. Among the type II patients, no differences in BMI or serum leptin levels were observed across the different prognostic categories (Table 4). Among the type I endometrial patients, only the levels of IL-6 among the proinflammatory cytokines were significantly higher in those with larger tumor size, higher nodal involvement, and distant metastases (stage IV). No significant differences in IL-6 serum levels were observed in the type II endometrial cancer patients, with respect to the prognostic categories T and N. Conversely, in the type II endometrial cancer patients, the IL-6 levels were significantly higher in stage IV patients in comparison with those who showed no evidence of distant metastases (Table 4). Among the type I endometrial cancer patients, the ROS levels were significantly higher in the patients with higher T, N, and FIGO stages (Table 5). Notably, among the type II endometrial cancer patients, the ROS levels were significantly higher in those with distant metastases (stage IV) than in those who showed no evidence of distant metastases. Among both the type I and II endometrial cancer patients, the GPx and SOD levels were not significantly different according to the T, N, G, and M states (Table 5).

### 3.4. Cross-Sectional Interrelationships between BMI, Leptin, Proinflammatory Cytokines, and Oxidative Stress Parameters in Endometrial Cancer Patients

We obtained a significant linear correlation of leptin with BMI in both the type I and II endometrial cancer patients. Moreover, both BMI and leptin were significantly positively related to IL-6 in the type I endometrioid endometrial cancer patients. Conversely, we did not find any significant relationship between BMI, leptin, and proinflammatory cytokines among the type II endometrial cancer patients (Table 6). We also observed a significant linear correlation between circulating ROS values and BMI, leptin, and IL-6 in the type I endometrial cancer patients. In the type II endometrial cancer patients, we did not observe any correlations between oxidative stress markers, BMI, or leptin, while we demonstrated a significant linear relationship between ROS and IL-6 (Table 6).

### 3.5. Correlation between Tumor Characteristics and BMI, Leptin, Proinflammatory Cytokine Levels, and Oxidative Stress Parameters among Type I and Type II Endometrial Cancer

By sub-dividing the patients according to endometrial cancer subtypes, we found that among the type I endometrial cancer patients, BMI, leptin, and IL-6 significantly correlated with T status, N status, and the presence of distant metastases (stage IV) (Table 7). Among the type II endometrial cancer patients, BMI and leptin did not correlate with any of the prognostic parameters, whereas a positive correlation between IL-6 and M status/stage was found. A significant positive correlation between the ROS levels and T and N status was found in the type I endometrial cancer patients. A significant positive correlation between ROS and the presence of distant metastases (stage IV) was found in both the type I and type II endometrial cancer patients (Table 7). No correlation was found between BMI, leptin, cytokines, and ROS with tumor grading in either the type I or II endometrial cancer patients. The multivariate regression analysis showed that BMI, leptin, and IL-6 were predictive for T, N, and M status in the type I endometrial cancer patients (Table 8). ROS were predictive of T and M status (Table 8).

## 4. Discussion

Epidemiological studies have shown a connection between excessive body weight and obesity and endometrial cancer [29], which has been ranked as the cancer with the highest risk per unit exposure among BMI-associated cancers in women [30]. The association between adipose tissue and estrogens is hypothesized to be one of the main pathways linking obesity and endometrial cancer pathogenesis, and the increased endometrial cancer risk among obese women has been strongly associated with circulating estrogen levels [31,32]. The unopposed estrogen hypothesis has historically been considered to be relevant for type I endometrioid cancer, which is typically recognized as being related to estrogen exposure and the obese phenotype [33,34]. Consistently, in this study, we found that the highest BMI was linked to type I endometrial cancer, and that BMI was correlated with T, N, and stage of disease in type I endometrial cancer patients, while no association was found in type II endometrial cancer patients. Therefore, BMI and obesity were correlated with poor standard prognostic factors, specifically in the type I subtype. The fact that BMI and obesity influence the prognosis of endometrial cancer has yet to be reported in the literature. In particular, two large population-based studies have reported that increased BMI is strongly associated with cancer-specific mortality from endometrial cancer [35,36]. Notably, and consistently with our results, a recent meta-analysis showed that when assessing the prognostic role of obesity according to endometrial cancer subtypes, the association between higher BMI and recurrence was significant among type I cases alone [37]. We did not find any correlation between BMI and tumor grading; this could surely have been influenced by the fact that we assessed high-grade endometrioid tumors together with type II non-endometrioid tumors, since they have been shown to share similar prognosis [22]. In this regard, a recent study by Faber et al. found that high-grade type I endometroid cancer patients, analyzed separately from low-grade type I endometrioid tumor patients, resembled type II tumors, regarding their association with BMI and obesity [33]. In this specific category of patients (high-grade ECs), an analysis of the whole tumor genomic profile becomes mandatory, as it has been highlighted that some of these patients show peculiar molecular alterations, mainly of the AKT/PI3K/mTOR pathways and the PIK3CA gene, which may open new therapeutic perspectives [38].

These results support the link between obesity and endometrial cancer, mediated by estrogens. However, the effects of obesity on endometrial cancer have been shown to be significant, even after adjusting for sex hormones [39], thereby suggesting the role of other adiposity-related pathways, which include insulin- and adipokine-mediated effects, the secretion of proinflammatory mediators from adipose tissue, and adiposity-associated oxidative stress [40,41].

Among adipokines, the most important member is leptin, whose circulating levels directly correlate with BMI and obesity [42,43]. The pathogenic role of leptin in obesity-linked malignancies, such as endometrial cancer, is complex. In cancer cell lines, leptin can be both mitogenic and antiapoptotic [44], and it is involved in carcinogenesis by initiating tumor cell proliferation, angiogenesis, and metastasis [43]. Leptin and its receptor (ObR) have been found in several cancers, including endometrial cancer [45]. Once bound to ObR, leptin exerts its effects by initiating the Janus kinase/signal transducers and the activator of the transcription pathway by phosphorylating tyrosine residues of ObR [46].

In this study, we found a correlation between leptin levels, BMI, and the standard prognostic factors for endometrial cancer. Specifically, increased leptin levels were found in type I endometrial cancer patients in comparison with type II endometrial cancer patients; furthermore, leptin was correlated with T, N, and M (stage of disease) in the type I endometrial cancer patients. Leptin was also an independent predictive factor of the main standard prognostic factors (T, N and M status) in type I cancer.

In the literature, leptin levels have been positively associated with an increased risk of endometrial cancer [47,48,49]. In a recent meta-analysis, Ellis et al. [8] showed that postmenopausal women with circulating leptin concentrations in the highest tertile had an increased endometrial cancer risk compared with women with concentrations in the lowest tertile. In the same study, BMI and diabetes appeared to affect the association between leptin levels and endometrial cancer risk. Moreover, in a large case-control study including postmenopausal women, Dallal et al. [29] found that increased leptin was associated with increased occurrence of endometrial cancer, and that, although leptin was correlated with BMI, endometrial cancer was more strongly associated with leptin than BMI, even after adjusting for estrogen levels. These data suggest that the leptin–BMI axis might increase endometrial cancer risk through mechanisms other than estrogen-driven proliferation. However, to date, limited studies have assessed the link between leptin and prognostic factors. One study conducted on 80 endometrial cancer patients showed that the expression levels of leptin and its receptor were associated with lymph node metastases and unfavorable prognosis in patients with endometrial cancer [50].

Our study also showed an association between obesity and inflammatory markers, and between the latter and standard cancer prognostic markers. Specifically, we found a positive relationship between IL-6 and BMI and leptin in type I endometrioid endometrial cancer, but not in type II. IL-6, in turn, was predictive of the tumor T, N, and stage. Indeed, adipose tissue-derived proinflammatory cytokines can trigger inflammation and mediate cancer progression. In endometrial cancer, both IL-6 and TNF-α have been reported to be overexpressed, and to play a role in cancer growth and metastasis, partly also by inducing ROS and subsequent DNA damage [51]. Peculiar histological structures, known as crown-like structures, characterized by dead or dying adipocytes surrounded by macrophages, are increased in the adipose tissue linked to obesity, and have been involved in the activation of nuclear factor-kB and the generation of a proinflammatory microenvironment. Concomitantly, they have been reported to play a role in the initiation and progression of endometrial cancer [52].

To date, some studies have assessed the role of proinflammatory cytokines in the risk of endometrial cancer. In detail, three large case studies demonstrated a relationship between the circulating levels of TNF-α, IL-6, IL-1α, and C-reactive protein (CRP) and elevated risk of endometrial cancer [53,54,55]. The association has been found to be largely dependent on the level of the patients’ adiposity [56]. In accordance with our results, these associations were significantly stronger in type I endometrial cancer [57,58]. Research evidence connects obesity with inflammation in endometrial cancer, and indicates that weight loss can reverse inflammation. Indeed, a paper demonstrated that, in a population of obese endometrial cancer patients, the circulating levels of CRP and TNF-α after bariatric surgery decreased to values closer to those observed in the control group, and that there was a significant correlation between a decline in BMI and lowering values of the inflammatory parameters CRP and TNFα [59].

Chronic systemic inflammation associated with obesity, and particularly IL-6 among cytokines, is also involved in the induction of insulin resistance and chronic hyper-insulinemia, which, in turn, are independent risk factors for the development of endometrial cancer. In our study, we did not investigate insulin levels, but we found that type I endometrial cancer patients, who were characterized by a high BMI, had a higher incidence of type II diabetes in comparison to type II endometrial cancer patients. Insulin can promote carcinogenesis via both the insulin receptor and insulin growth factor (IGF), whose circulating levels increase as result of the insulin-mediated decrease in IGFBP1 and IGFBP2. Both insulin and IGF induce a multitude of tumor-promoting mechanisms involved in cell proliferation, anti-apoptosis, angiogenesis, and lymphangiogenesis [60]. These effects are mediated by the activation of the PI3K/AKT/mTOR pathway and the Ras/Raf/MEK mitogen-activated protein kinase (MAPK) pathway. Of note, hyperactivation of mTOR signaling has been observed in experimental models of endometrial cancer associated with obesity and in a large fraction of endometrial tissues from obese endometrial cancer patients, compared with non-obese patients [61]. Overactivation of the PI3K/AKT/mTOR pathway, due to the loss of the tumor suppressor gene phosphatase and tensin homolog (PTEN), has been observed in more than 40% of type I endometrial cancers. Interestingly, the PI3K/AKT/mTOR pathway mediates signaling from leptin and IL-6 [62], which, in our study, have been found to be significantly elevated in type I endometrial cancer patients and positively correlated with tumor prognostic factors. Thus, this pathway could be a potential target in obesity-related endometrial cancer.

A crucial intracellular pathway involved in the development of insulin resistance and diabetes is the 5′ adenosine monophosphate-activated protein kinase (AMPK) pathway. The metabolic syndrome phenotype with associated inflammation, hyperglycemia, and hyperinsulinemia is associated with reduced AMPK activity, which contributes to the development of diabetes and its complication. In turn, AMPK activation can reverse insulin resistance, hypertriglyceridemia, diabetes, oxidative stress, and inflammation [63]. Reduced activity of AMPK has been observed in endometrial cancer cells, and has been associated with cancer cell proliferation, invasion, and progression [64]. Therefore, activating AMPK by specific agents may offer a promising therapeutic target for obesity-related endometrial cancer at different levels.

Obesity has also been associated with increased levels of ROS, which contribute to tumor promotion. Moreover, hyperglycemia, along with elevated free fatty acid levels, induces ROS production, which additively provokes mitochondrial and nucleolar DNA damage [65]. Additionally, proinflammatory cytokines secreted by adipose tissue may induce the production of ROS. In this study, we found that ROS levels were significantly higher in endometrial cancer patients than in controls, and were higher in type I cancers than type II cancers. ROS correlated with BMI, leptin, and IL-6, as well as with the main standard prognostic factors T, N, and stage. Strikingly, the correlation between ROS and IL-6, and between ROS and stage of disease, has been found in both type I and II endometrial cancer patients, thus substantiating the evidence showing that metastatic disease and chronic inflammation are associated with ROS independently from adiposity. Consistently with our findings, another paper reported that endometrial cancer patients had increased levels of oxidative stress markers, in comparison to controls, and that they were associated with increased values of circulating TNF-α, IL-6, and CRP [18].

Knowing the role of obesity-related biological pathways and mediators in the pathogenesis and prognosis of endometrioid endometrial malignancies offers new perspectives and therapeutic strategies that have not been explored to date. A combined approach, comprising aromatase inhibitors combined with antidiabetic, anti-inflammatory, and antioxidant drugs, plus progestins administered locally through an intrauterine device [66], may constitute a new targeted therapeutic approach, particularly in women that, by virtue of obesity, may also have an increased risk of surgical complications and, therefore, could not receive optimal surgery. The contribution of targeting these pathways in the context of adjuvant treatment is no less important, where the information available is limited and scarcely homogeneous.

Among anti-diabetic drugs, metformin showed some promising evidence of ameliorating the prognosis of obese endometrial cancer patients. Metformin is able to act on multiple mechanisms involved in the association between adiposity and EC pathogenesis; it reduces insulin receptor activity and leads to the activation of AMPK, affecting the mTOR pathway [67]. Other drugs, such as thiazolidinediones and several natural agents, may activate AMPK [63]. Additionally, the adiposity-related upregulation of the mTOR and VEGF pathways may support the use of their inhibitors as promising therapeutic options [61]. Notably, since the increased expression of aromatase in type I endometrial cancer has been associated with the expression of cyclo-oxygenase (COX)-2, the inhibition of prostaglandin 2, i.e., by aspirin or COX-2 inhibitors, may result in reduced estrogen biosynthesis, and could provide a potential preventive/therapeutic target. In this regard, it is worthy to note that the non-steroidal anti-inflammatory agent acetylsalicylic acid has been associated with decreased incidence and mortality of endometrial cancers [68,69].

The key components of such an integrated approach are diet and physical activity to reduce body weight and obesity. In this regard, it has been demonstrated that the dietary intake of fish, omega 3, and fiber, as well as moderately intense aerobic exercise, can directly modulate adiponectin/leptin levels [70,71]. Notably, a recent paper showed that weight loss combined with the administration of intrauterine progestin levonorgestrel obtained a high percentage of patients with a pathological complete response in patients with endometrial adenocarcinoma and endometrial hyperplasia with atypia [66].

## 5. Conclusions

Our study demonstrates that weight gain and adiposity-related adipokines, inflammation, and oxidative stress are involved in influencing the prognosis of type I endometrial cancer. The evidence coming from observational studies, such as the current work, is expected to be translated into clinical practice to develop targeted therapies that are potentially effective in improving the prognosis of endometrial cancer in the population of overweight/obese patients [72]. Moreover, the identification of circulating biomarkers, such as leptin, proinflammatory cytokines, and ROS, could be very useful in patients’ risk stratification, and in identifying potential candidates for specific tailored treatments. Large and well-designed randomized studies are warranted to clinically test the proposed therapeutic approaches.

## Figures and Tables

**Table 1 cancers-14-00268-t001:** Baseline anthropometric and laboratory characteristics of endometrial cancer patients and healthy controls.

Parameters	Endometrial Cancer Patients *N* = 305
Age, years (mean ± SD)	65.3 ± 8.6
Weight, kg (mean ± SD)	63.1 ± 10.5
Height, cm (mean ± SD)	157.5 ± 7.5
BMI	27.7 ± 7.8
Menopausal status, No. (%)	
Premenopausal	46 (15)
Postmenopausal	216 (71)
Perimenopausal	43 (14)
Tumor size (pT), No. (%)	
T1	189 (62)
T2	77 (25)
T3	23 (8)
T4	16 (5)
Grading (G), No. (%)	
G1	124 (40.7)
G2	92 (30.1)
G3	89 (29.2)
Nodal Status (pN), No. (%)	
N0	189 (61.9)
N1	85 (27.9)
N2	20 (6.6)
N3	11 (3.6)
Stage of disease (FIGO), No. (%)	
I	152 (49.8)
II	100 (32.9)
III	37 (12.1)
IV	16 (5.2)
Leptin, ng/mL (mean ± SD)	25.5 ± 12.3
IL-6, pg/mL (mean ± SD)	20.8 ± 7.9
TNF-alpha, pg/mL (mean ± SD)	23.9 ± 8.7
ROS, FORT U (mean ± SD)	355 ± 169
GPx, U/mL (mean ± SD)	7969 ± 4580
SOD, U/L (mean ± SD)	100.5 ± 65

Abbreviations: BMI, body mass index; SD, standard deviation.

**Table 2 cancers-14-00268-t002:** Baseline clinical and tumor characteristics of endometrial cancer patients according to tumor type.

Parameter	Type I Cancer (G1-2 Endometrioid Histology) No. (%)	Type II Cancer (Non-Endometrioid and G3 Endometrioid Histology) No. (%)	*p*-Value
Enrolled patients	203	102	
BMI, kg/m^2^ (mean ± SD)	28.7 ± 4.6	24.6 ± 3.9	**0.0001 (95% CI –4.16 to –1.44)**
BMI categories			
<18.5	5 (2)	2 (2)	**0.0018**
18.5–24.9	80 (39)	75 (74)	
25–29.9	66 (33)	20 (20)	
≥30	53 (26)	4 (4)	
History of diabetes			
Yes	132 (65)	27 (26)	<0.0001
No	72 (35)	74 (73)	
Histology			NA
Endometrioid	203 (100)	51 (50)	
Serous adenocarcinoma	NA	22 (21)	
Clear cell	NA	8 (8)	
Mucinous	NA	7 (7)	
Mixed cell	NA	14 (14)	
Tumor size (T)			
T1	142 (70)	47 (46)	0.0746
T2	43 (21)	34 (33)	
T3	10 (5)	13 (13)	
T4	8 (4)	8 (8)	
Grading (G)			
G1	122 (60)	2 (2)	<0.0001
G2	81 (40)	11 (11)	
G3	0	89 (87)	
Nodal Status (N)			
N0	130 (64)	59 (58)	0.3538
N1	49 (24.2)	36 (35)	
N2	15 (7.4)	5 (5)	
N3	9 (4.4)	2 (2)	
Stage of disease (FIGO)			
I	112 (55)	40 (39)	0.3441
II	59 (29)	41 (40)	
III	24 (12)	13 (13)	
IV	8 (4)	8 (8)	

*p*-value was calculated by Student’s *t*-test for comparison between means and chi-squared test for comparison between proportions. *p*-value < 0.05 is considered significant (bold value). Abbreviations: SD, standard deviation.

**Table 3 cancers-14-00268-t003:** Baseline laboratory parameters in endometrial cancer patients divided according to tumor type.

Parameter	Type I Cancer (G1–G2 Endometrioid) *N* = 203	Type II Cancer (Non-Endometrioid and G3 Endometrioid) *N* = 102	*p*-Value	95% CI
Leptin, ng/mL (mean ± SD)	58.5 ± 18	18.3 ± 9.6	**<0.0001**	**−43.18 to −24.19**
IL-6, pg/mL (mean ± SD)	24.1 ± 11.5	18.3 ± 10.7	**<0.0001**	**−28.1 to −11.9**
TNF-α, pg/mL (mean ± SD)	18.7 ± 9.8	22.1 ± 10.2	0.1650	−11.76 to 2.678
ROS, FORT U (mean ± SD)	497 ± 152	397 ± 98	**<0.0001**	**−89.98 to −35.4**
GPx, U/mL (mean ± SD)	8165 ± 2590	7464 ± 1228	0.1357	−25.16 to 15.3
SOD, U/L (mean ± SD)	108 ± 49	117 ± 67	0.3272	−28.7 to 16.7

*p*-value was calculated by Student’s *t*-test. *p*-value < 0.05 is considered significant (bold value). Abbreviations: SD, standard deviation; IL, interleukin; TNF, tumor necrosis factor; ROS, reactive oxygen species; GPx, glutathione peroxidase; SOD, superoxide dismutase.

**Table 4 cancers-14-00268-t004:** Evaluation of BMI, leptin, and proinflammatory cytokines according to prognostic categories among endometrial cancer subtypes.

Parameter	Type I Cancer (G1–G2 Endometrioid) *N* = 203				Type II Cancer (Non-Endometrioid and G3 Endometrioid) *N* = 102			
	BMI (kg/m^2^)	Leptin (ng/mL)	IL-6 (pg/mL)	TNFa (pg/mL)	BMI (kg/m^2^)	Leptin (ng/mL)	IL-6 (pg/mL)	TNFa (pg/mL)
Tumor size (T)								
T1	23.1 ± 3.6	17.5 ± 6.7	8.6 ± 2.8	16.4 ± 3.2	22.1 ± 4.2	15.3 ± 4.5	11.7 ± 4.2	22.6 ± 10.9
T2	24.6 ± 1.3	22.9 ± 16.2	11.9 ± 3.6	20.6 ± 10.3	23.5 ± 6.1	22.1 ± 5.3	19.2 ± 10.7	20.9 ± 9.5
T3	25.2 ± 3.4	38.5 ± 23.5	19.2 ± 4.5	18.1 ± 9.6	24.1 ± 6.2	23.9 ± 11.7	23.8 ± 12.3	21.4 ± 11.2
T4	27.3 ± 4.2 *	45.7 ± 21.7 *	25.9 ± 8.7 *	21.5 ± 12.7	23.9 ± 8.2	22.8 ± 10.6	24.6 ± 10.9	25.3 ± 6.7
Grading (G)								
G1	26.4 ± 4.1	29.9 ± 19.7	16 ± 12	20.1 ± 11	21.7 ± 4.5	16.7 ± 13	19.7 ± 5	17.6 ± 6.7
G2	25.5 ± 2.7	27.5 ± 18.4	22 ± 10	15.5 ± 5.4	22.4 ± 6.5	21.8 ± 11.2	14.8 ± 7.8	11.4 ± 9.8
G3	NA	NA	NA	22.7 ± 9.8	22.8 ± 5.4	21.7 ± 14.1	19.5 ± 7.6	25.7 ± 7.8
Nodal Status (N)								
N0	23.8 ± 5.3	22 ± 14.5	9.2 ± 5.6	18 ± 5.7	23.6 ± 4.4	18.4 ± 6.2	7.7 ± 5.4	19.8 ± 10.3
N1	24.7 ± 5.6	36 ± 18.7	20.3 ± 10.6	19 ± 8.2	24.5 ± 5.8	23.9 ± 13.2	18.2 ± 9.6	20.1 ± 9.7
N2	26.4 ± 4.1	45.2 ± 23	29.1 ± 13.7	23 ± 10.4	23.5 ± 4.2	22 ± 8.7	21 ± 10.5	22 ± 11.5
N3	27.9 ± 2.8 *	49.8 ± 21.1 *	36.3 ± 16.8 *	26.3 ± 11.5	22.1 ± 5.3	16.5 ± 9.6	23.1 ± 7.7	23.2 ± 7.6
Stage of disease								
I–III	23.9 ± 6.2	20.4 ± 8.5	14.6 ± 4.1	19 ± 6.5	22.8 ± 7.6	19.7 ± 9.8	13.4 ± 5.3	18 ± 7.8
IV	34 ± 11.6 *	56.2 ± 21.1 *	44.2 ± 10.5 *	26.7 ± 11.9	23.5 ± 6.8	20.5 ± 13.6	30.5 ± 14.3 *	24.8 ± 11.1

Data are reported as mean ± SD. * *p* < 0.05 as calculated by ANOVA test. Abbreviations: BMI, body mass index; IL, interleukin; TNF, tumor necrosis factor.

**Table 5 cancers-14-00268-t005:** Evaluation of oxidative stress parameters according to endometrial cancer subtypes.

Parameter	Type I Cancer (G1–G2 Endometrioid) *N* = 203			Type II Cancer (Non-Endometrioid and G3 Endometrioid) *N* = 102		
	ROS (FORT U)	GPx (U/L)	SOD (U/mL)	ROS (FORT U)	GPx (U/L)	SOD (U/mL)
Tumor size (T)						
T1	331 ± 47	9471 ± 1230	156 ± 45	278 ± 46	9906 ± 473	98 ± 41
T2	386 ± 42	9323 ± 1543	179 ± 41	369 ± 51	9452 ± 510	88 ± 23
T3	415 ± 124	8579 ± 2145	148 ± 44	379 ± 74	8530 ± 865	85 ± 52
T4	448 ± 101 *	8185 ± 1274	105 ± 89 *	427 ± 89	8480 ± 1186	96 ± 48
Grading (G)						
G1	386 ± 74	8299 ± 1197	160 ± 12	387 ± 45	9167 ± 1300	97 ± 33
G2	368 ± 97	8275 ± 1184	122 ± 50	324 ± 65	9218 ± 1120	98 ± 38
G3	NA	NA	NA	409 ± 55	8741 ± 1450	154 ± 97
Nodal Status (N)						
N0	342 ± 43	9123 ± 1455	117 ± 42	388 ± 56	9076 ± 1308	91 ± 32
N1	378 ± 55	9264 ± 1590	129 ± 37	379 ± 72	8539 ± 1143	92 ± 31
N2	446 ± 116	8265 ± 1210	109 ± 46	429 ± 57	8480 ± 1430	81 ± 51
N3	489 ± 133 *	7758 ± 1113	111 ± 57	438 ± 84	8175 ± 1560	82 ± 46
Stage of disease						
I–III	377 ± 51	9056 ± 1060	118 ± 53	327 ± 67	9364 ± 1080	112 ± 46
IV	476 ± 115 *	8465 ± 1153	97 ± 38	416 ± 54 *	8756 ± 1260	109 ± 43

Data are reported as mean ± SD. * *p* < 0.05 as calculated by ANOVA test. Abbreviations: ROS, reactive oxygen species; SOD, superoxide dismutase; GPx, glutathione peroxidase.

**Table 6 cancers-14-00268-t006:** Cross-sectional interrelationship between BMI, leptin, proinflammatory cytokines, and oxidative stress parameters.

Parameter	BMI		Leptin		IL-6		TNF-α		ROS		GPx	
	r	*p*	r	*p*	r	*p*	r	*p*	r	*p*	r	*p*
Type I cancer (G1–G2 endometrioid)												
BMI												
Leptin	0.874	**<0.001**										
IL-6	0.540	**<0.001**	0.895	**<0.001**								
TNF-α	0.175	0.072	0.331	0.087	0.419	**0.006**						
ROS	0.458	**0.010**	0.655	**<0.001**	0.844	**<0.001**	0.276	0.089				
GPx	0.109	0.354	0.089	0.680	−0.547	**0.006**	−0.228	0.473	−0.493	**0.007**		
SOD	0.053	0.918	0.154	0.675	−0.546	**0.021**	−0.364	0.073	−0.430	**0.010**	0.280	0.105
Type II cancer (non-endometrioid and G3 endometrioid)												
BMI												
Leptin	0.876	**<0.001**										
IL-6	−0.144	0.394	0.168	0.210								
TNF-α	0.165	0.138	−0.392	0.328	0.410	0.060						
ROS	−0.250	0.129	0.178	0.249	0.695	**0.003**	0.109	0.570				
GPx	−0.198	0.215	−0.071	0.508	−0.234	0.231	−0.197	0.144	0.105	0.684		
SOD	−0.157	0.453	−0.163	0.218	−0.305	0.178	0.239	0.106	−0.760	**0.004**	0.348	0.075

Abbreviations: r, correlation factor; EC, endometrial cancer; BMI, body mass index; IL, interleukin; TNF, tumor necrosis factor; ROS, reactive oxygen species; GPx, glutathione peroxidase; SOD, superoxide dismutase. *p*-value was calculated by Pearson’s or Spearman’s correlation analysis and was considered significant if it was less than 0.05 (bold value).

**Table 7 cancers-14-00268-t007:** Correlation between BMI, leptin, proinflammatory cytokines, and oxidative stress parameters within tumor prognostic characteristics.

Parameter	BMI		Leptin		IL-6		TNF-α		ROS		GPx		SOD	
	r	*p*	r	*p*	r	*p*	r	*p*	r	*p*	r	*p*		
Type I cancer (G1–G2 endometrioid)														
T	0.568	**0.019**	0.550	**0.011**	0.718	**<0.001**	0.107	0.154	0.436	**0.010**	−0.167	0.270	−0.099	0.644
N	0.260	0.108	0.486	**0.015**	0.683	**0.001**	0.165	0.120	0.458	**0.008**	−0.196	0.215	0.086	0.580
G	0.289	0.102	0.263	0.273	0.190	0.298	0.078	0.809	0.079	0.746	0.376	0.061	0.298	0.091
Stage	0.455	**0.019**	0.765	**<0.001**	0.823	**<0.001**	0.187	0.680	0.759	**<0.001**	−0.275	0.081	−0.287	0.087
Type II cancer (non-endometrioid and G3 endometrioid)														
T	0.056	0.845	0.257	0.293	0.295	0.159	0.186	0.374	0.395	0.078	−0.276	0.143	−0.081	0.796
N	0.285	0.176	0.156	0.452	0.297	0.168	0.176	0.398	0.297	0.081	−0.246	0.228	−0.099	0.608
G	−0.299	0.108	−0.276	0.094	0.191	0.475	−0.198	0.454	0.392	0.128	−0.085	0.879	−0.226	0.245
Stage	0.099	0.687	0.275	0.104	0.506	**0.025**	0.176	0.184	0.678	**0.002**	−0.156	0.398	−0.085	0.615

Abbreviations: r, correlation factor; EC, endometrial cancer; BMI, body mass index; IL, interleukin; TNF, tumor necrosis factor; ROS, reactive oxygen species; GPx, glutathione peroxidase; SOD, superoxide dismutase. *p*-value was calculated by Pearson’s or Spearman’s correlation analysis and was considered significant if it was less than 0.05 (bold value).

**Table 8 cancers-14-00268-t008:** Multivariate regression analysis of BMI, leptin, proinflammatory cytokines, and oxidative stress parameters with regard to standard tumor prognostic factors in type I endometrial cancer patients.

Parameter	T		N		Stage	
	Beta Coefficient	*p*	Beta Coefficient	*p*	Beta Coefficient	*p*
Type I EC						
BMI (kg/m^2^)	0.849	**<0.001**	0.544	**0.033**	0.515	**0.018**
Leptin (ng/L)	0.619	**0.025**	0.464	**0.028**	0.454	**0.032**
IL-6 (pg/mL)	0.717	**0.014**	0.986	**<0.001**	0.698	**0.002**
ROS (FORT U)	0.558	**0.021**	0.398	0.058	0.586	**0.021**

Abbreviations: EC, endometrial cancer; BMI, body mass index; IL, interleukin; TNF, tumor necrosis factor; ROS, reactive oxygen species. *p*-value was considered significant if it was less than 0.05 (bold value).

## Data Availability

Original clinical, laboratory and instrumental data can be found in the patients’ charts archived at the Department of Obstetrics and Gynecology, A. Businco Hospital, RNAS G. Brotzu, and are available on request from the corresponding author. Data are also available from Sardegna Ricerche and are available at https://smec.regione.sardegna.it/ with the permission of Sardegna Ricerche (www.sardegnaricerche.it).

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
