# Peer review of "Correlation of Leptin, Proinflammatory Cytokines and Oxidative Stress with Tumor Size and Disease Stage of Endometrioid (Type I) Endometrial Cancer and Review of the Underlying Mechanisms"

_cancers, 2022, doi:10.3390/cancers14020268_

Round 1

Reviewer 1 Report

This work is the interesting and comprehensive summary for the understanding of the treated topic. The abstract and introduction provide adequate information on the discussion of the following paragraphs

.  the manuscript requires further review by the authors. 

Author Response

Point-by-point reply to Reviewer 1

This work is the interesting and comprehensive summary for the understanding of the treated topic. The abstract and introduction provide adequate information on the discussion of the following paragraphs

.  the manuscript requires further review by the authors. 

Reply: I appreciate the positive comments of the reviewer. I have revised the study design and results and I have checked the English style and grammar and revised it according to your suggestions in the review report form.

Reviewer 2 Report

              The prospective study by Dr. Madeddu and co-workers is an interesting and well executed investigation on interrelationships of clinical risk factors and anatomopathological prognosticators for endometrial cancer (EC) with adipose tissue hormonal and metabolic products, this done at an Italian referral center over a 7.5-year period. The study is quite well conducted and the literature selection attractive. However, I have a major methodological issue regarding the study design (see below) that affects the whole outcome. Hopefully, the study can be rescued by adequate recalculations.  

              Major issues:  

- the Authors relied on an old type I and type II EC division which is based on EC histotypes alone; this outdated and clinically improper Bokhman’s classification from 1983 has been seriously criticized [see for example: Setiawan, V.W.; Yang, H.P.; Pike, M.C.; McCann, S.E.; Yu, H.; Xiang, Y.B.; Wolk, A.; Wentzensen, N.; Weiss, N.S.; Webb, P.M.; et al. Type i and II endometrial cancers: Have they different risk factors? J. Clin. Oncol. 2013, 31, 2607–2618.] since over time the survival of endometrioid EC G3 patients was found as bad as of type II patients; now your study does not properly distinguish good prognosis patients from bad prognosis patients ! ;  now type II EC includes endometrioid G3 cases !

- there is no mention in the text which FIGO classification (and this evolved) was used for staging whereas this is a study tool; 

- type II patients were not characterized for their histotypes.

              Minor – many stylistical and grammatical impefections:  

- Abstract, is: ‘ Endometrioid endometrial cancer is associated to increased BMI … ‘; ‘to’ is ill-chosen;  

- Abstract, the phrase ‘Our study demonstrates that weight gain, and adiposity-related adi-44 pokines, inflammation, and oxidative stress correlate with prognostic factor of endometrioid endo-45 metrial cancer.’ is stylistically imperfect;  

- lines 48-49: is: ‘ … in the metastatic (…) setting.’ - ?

- line 87: correct ‘view’ to viewed’; 

- lines 91-92, please reference a powerful statement: ‘Indeed, IL-6 was found to be over-91 expressed in the stroma of endometrial cancer cells’; 

- lines 98-100, please reference a powerful statement: ‘Moreover, they (ROS) are also able to induce the activation of Akt/PI3K/mTOR signaling involved in endometrial cancer (EC) proliferation and progression.’; 

- line 115 and further, the exclusion criteria are incomplete, for instance, the co-existance of any other malignancy should be included;

- line 147, is the word ‘calculated’ proper ?

- line 409, is: ‘ … involved in conditioning the prognosis of Type I endometrial cancer.’; ‘conditioning’ sounds suboptimal in this setting; 

- ‘5. Conclusions’ is exceedingly long and provides new references (!) and contains elements of Discussion;  

- References: many titles are unnecessarily capitalized;

etc.

Author Response

Point-by-point reply to Reviewers’ comments

Reviewer 2

The prospective study by Dr. Madeddu and co-workers is an interesting and well executed investigation on interrelationships of clinical risk factors and anatomopathological prognosticators for endometrial cancer (EC) with adipose tissue hormonal and metabolic products, this done at an Italian referral center over a 7.5-year period. The study is quite well conducted and the literature selection attractive. However, I have a major methodological issue regarding the study design (see below) that affects the whole outcome. Hopefully, the study can be rescued by adequate recalculations. 

Major issues: 

- the Authors relied on an old type I and type II EC division which is based on EC histotypes alone; this outdated and clinically improper Bokhman’s classification from 1983 has been seriously criticized [see for example: Setiawan, V.W.; Yang, H.P.; Pike, M.C.; McCann, S.E.; Yu, H.; Xiang, Y.B.; Wolk, A.; Wentzensen, N.; Weiss, N.S.; Webb, P.M.; et al. Type I and II endometrial cancers: Have they different risk factors? J. Clin. Oncol. 2013, 31, 2607–2618.] since over time the survival of endometrioid EC G3 patients was found as bad as of type II patients; now your study does not properly distinguish good prognosis patients from bad prognosis patients ! ;  now type II EC includes endometrioid G3 cases !

Reply: Thank you for your very punctual and useful comment. I appreciate them. Indeed, in the first version of the manuscript I have included among type I endometrioid cancers also G3 endometrioid cases in accordance with the classification reported in a review in NEJM published in 2020 (see ref 24 of the revised version), where grade 3 endometrial cancers were considered among together with low-grade endometrioid cancers. However, as you correctly have observed the classification in type I and II according to Bookman is obsolete and I am convinced, as we have commented in the discussion of the first version of the manuscript and as well clarified by you, that “high grade type I tumors resemble type II tumors……”. At this regard we can highlight how, in the results of the first version submitted, BMI and leptin did not correlate with grading also in type I endometrial cancers.  Therefore, according to your suggestion and the evidence reported in literature, including the reference suggested by you (J Clin Oncol 2013; 31:2607-2618; Gynecol Oncol 2012; 124-15-20) in the revised version we have included grade 3 endometrioid cases into type II EC group. I hope that this new division among type I (grade 1-2 endometrioid) and type II (non-endometrioid and G3 endometrioid) cancers could more properly distinguish good and poor prognosis patients. I have clarified this point firstly in the aim of the study (lines 112-115 of the revised version), and I have specified these criteria in the Methods Section at lines 136-142 of the revised version. Therefore, tables and data have been revised accordingly.

- there is no mention in the text which FIGO classification (and this evolved) was used for staging whereas this is a study tool;

Reply: I have specified that we used the revised 2009 FIGO staging system for endometrial cancer (see line 143 of the revised version).

- type II patients were not characterized for their histotypes.

Reply: I have added the histotypes of type II patients (see line 138-139 and Table 1 of the revised version).

Minor – many stylistic and grammatical imperfections: 

Reply: I have checked again English style and grammar and corrected the errors.

- Abstract, is: ‘ Endometrioid endometrial cancer is associated to increased BMI … ‘; ‘to’ is ill-chosen; 

Reply: I have corrected “associated to” into “associated with”

- Abstract, the phrase ‘Our study demonstrates that weight gain, and adiposity-related adipokines, inflammation, and oxidative stress correlate with prognostic factor of endometrioid endometrial cancer.’ is stylistically imperfect; 

Reply: I have changed the sentence as follows: “Our study demonstrates that weight gain, adiposity-related adipokines, inflammation, and oxidative stress correlate with prognostic factor of endometrioid endometrial cancer.”

- lines 48-49: is: ‘ … in the metastatic (…) setting.’ - ?

Reply: I have corrected the text as follows: “both in the advanced disease and in the adjuvant setting.”

- line 87: correct ‘view’ to viewed’;

Reply: thank you. I have corrected “view” into “viewed” as indicated by you

- lines 91-92, please reference a powerful statement: ‘Indeed, IL-6 was found to be overexpressed in the stroma of endometrial cancer cells’;

Reply: I have added the appropriate references for the statement as indicated by you (lines 93-94 of the revised version).

- lines 98-100, please reference a powerful statement: ‘Moreover, they (ROS) are also able to induce the activation of Akt/PI3K/mTOR signaling involved in endometrial cancer (EC) proliferation and progression.’

Reply: I have added the reference that support the statement (lines 100-102 of the revised version).

- line 115 and further, the exclusion criteria are incomplete, for instance, the coexistance of any other malignancy should be included;

Reply: I have included among exclusion criteria the coexistence of any other malignancy (see line 122 of the revised version)

- line 147, is the word ‘calculated’ proper ?

Reply:  I have corrected the term “calculated” into “analyzed”

- line 409, is: ‘ … involved in conditioning the prognosis of Type I endometrial cancer.’; ‘conditioning’ sounds suboptimal in this setting;

Reply: Thank you for your comment. I have changed the term “conditioning” into “influencing”.

- ‘5. Conclusions’ is exceedingly long and provides new references (!) and contains elements of Discussion;

Reply: In accordance with your comment, I have shortened the conclusion and moved several sentences to Discussion. I have left in the Conclusions the following sentences and moved the other ones at the end of the Discussion: “Our study demonstrates that weight gain, and adiposity-related adipokines, inflammation, and oxidative stress are involved in influencing the prognosis of Type I endometrial cancer. The evidence coming from observational studies, such as the cur-rent work, is expected to be translated into clinical practice to develop targeted therapies that are potentially effective in improving the prognosis of endometrial cancer in the population of overweight/obese patients [71]. Moreover, the identification of bi-omarkers in peripheral serum, such as leptin, proinflammatory cytokines, and ROS, could be very useful in patients’ risk stratification and in identifying potential candi-dates for specific tailored treatments. Large and well-designed randomized studies are warranted to clinically test the proposed therapeutic approaches.”

- References: many titles are unnecessarily capitalized

Reply: I have corrected the titles and revised their style.

Reviewer 3 Report

Interesting paper about Correlation of leptin, proinflammatory cytokines and oxidative stress with tumor size and disease stage of endometrioid (Type I) endometrial cancer and review of the underlying mechanisms. Impressive Results part.

Please check bellow my few minor suggestions:

Please check to use just one type of English (British or American), not both.

Please check the entire manuscript related the supportive references. Some longer paragraphs must be referenced. (i.e., the paragraph between L55-66 is not referenced at all. I suggest  to check and refer to https://doi.org/10.3390/medicina57090945

Table 1., Table 3, etc. As the unit of measure for volume in your research, please replace ml with mL (as Litter being the international unit of measure for volume) (i.e. Table 3, 3rd row). Please check/revise the entire manuscript in this regard.

4. Discussion. Please detail the role that the AMPK pathway plays in the generation of metabolic alterations and oncogenic activities and potential way to inhibit AMPK activation in these patinets. AMPK is an important pathway in insuslin-rezistance and inflammation, and discussions can be further improved. In this regard I suggest to check: https://www.springermedizin.de/a-spotlight-on-underlying-the-mechanism-of-ampk-in-diabetes-comp/19399486

5.Conclusions section must be shortened. It is extremely long. This section must briefly present, usually in a single /unique paragraph the main findings of your research. Some information are in duplicate with other parts of the manuscript - please remove them. Moreover, a good part of this section can be moved to the Discussion.

Author Response

Point-by-point reply to Reviewer 3

Interesting paper about Correlation of leptin, proinflammatory cytokines and oxidative stress with tumor size and disease stage of endometrioid (Type I) endometrial cancer and review of the underlying mechanisms. Impressive Results part.

Please check bellow my few minor suggestions:

-Please check to use just one type of English (British or American), not both.

Reply: I have checked English and uniformed the text in American English.

-Please check the entire manuscript related the supportive references. Some longer paragraphs must be referenced. (i.e., the paragraph between L55-66 is not referenced at all. I suggest to check and refer to https://doi.org/10.3390/medicina57090945

Reply: I have checked and added more references, including also that suggested by you.

Table 1., Table 3, etc. As the unit of measure for volume in your research, please replace ml with mL (as Liter being the international unit of measure for volume) (i.e. Table 3, 3rd row). Please check/revise the entire manuscript in this regard.

Reply: I have checked the tables and the entire manuscript and I have replaced ml with mL.

  1. Discussion. Please detail the role that the AMPK pathway plays in the generation of metabolic alterations and oncogenic activities and potential way to inhibit AMPK activation in these patients. AMPK is an important pathway in insulin-resistance and inflammation, and discussions can be further improved. In this regard I suggest to check: https://www.springermedizin.de/a-spotlight-on-underlying-the-mechanism-of-ampk-in-diabetes-comp/19399486

Reply: Thank you for your suggestion. I have added in the discussion some sentences on the role of AMPK in the generation of metabolic alterations and diabetes as well as on its involvement in promoting proliferation and invasion of endometrial cancer cells. Moreover, I have added a sentence on the different therapeutic approaches able to influence AMPK activation and thus be potentially useful for the treatment of endometrial cancer related to obesity, diabetes, and insulin-resistance. See lines 420-429 and lines 458-463 of the revised version.

5.Conclusions section must be shortened. It is extremely long. This section must briefly present, usually in a single /unique paragraph the main findings of your research. Some information are in duplicate with other parts of the manuscript - please remove them. Moreover, a good part of this section can be moved to the Discussion.

Reply: I have shortened the Conclusions section to the following sentences “Our study demonstrates that weight gain, and adiposity-related adipokines, inflammation, and oxidative stress are involved in influencing the prognosis of Type I endometrial cancer. The evidence coming from observational studies, such as the cur-rent work, is expected to be translated into clinical practice to develop targeted therapies that are potentially effective in improving the prognosis of endometrial cancer in the population of overweight/obese patients [71]. Moreover, the identification of biomarkers in peripheral serum, such as leptin, proinflammatory cytokines, and ROS, could be very useful in patients’ risk stratification and in identifying potential candidates for specific tailored treatments. Large and well-designed randomized studies are warranted to clinically test the proposed therapeutic approaches”. The other parts have been moved to the end of the Discussion, trying to avoid duplicate.

Round 2

Reviewer 2 Report

              The Authors have complied to the major points of criticism regarding their first version of the MS. Apart from a rather long, if not lengthy, Discussion, I have only minor issues to tackle, of which the missing methodology for Ki-67 is the most important. Smoothing the text by a native English speaker would be a nice addition.  

              My current indications include:  

- line 77, pls remove the second (unnecessary) hyphen;  

- lines 101 and 407: arrange for explaining mTOR with the first use of the abbreviation;

- line 113, pls change ‘the increasing reports that’ to ‘several important reports’; 

- line 122, change ‘coexisting any’ to ‘any coexisting’;  

- line 128, is: ‘2.2. Measurement’; should be plural;

- line 154, add plural to picogram; 

- lines 181-183, is: ‘Table 1 and Table 2 report the baseline patients’ clinical and tumor character-istics of endometrial cancer globally and grouped according to cancer subtypes (Type I and II cancer), respectively.’; a better wording would be: ‘Tables 1 and 2 report the patients’ clinical characteristics and studied tumors characteristics, both for the whole group and divided into Type I and II cancer subtypes.’; 

- line 186, of out the sudden ‘Ki-67 expression’ appears; Ki-67 evaluation has not been given in the Methods section;  

- Table 3: change ‘(G1 endometrioid) No. 203’ to ‘(G1-2 endometrioid; N = 203) ’, 'No. 102' to 'N = 102', and ‘P value’ to ‘P-value’; 

- line 472, „Pointedly’’ is a rarely used word in English language;

- some references have their titles unnecessarily capitalized; some references unnecessarily provide doi numbers.  

Thank you.

Author Response

Point-by-point reply to Reviewers’ comments ROUND 2

Reviewer 2

Comments and Suggestions for Authors

The Authors have complied to the major points of criticism regarding their first version of the MS. Apart from a rather long, if not lengthy, Discussion, I have only minor issues to tackle, of which the missing methodology for Ki-67 is the most important. Smoothing the text by a native English speaker would be a nice addition. 

Reply: Thank you for your positive comments about the revision of the major criticisms of the first version. I am pleased that they addressed your requirements. As regard the length of Discussion I have shortened it, compatibly with the indication furnished by the Reviewer 3 that required to add an entire paragraph on the AMPK role. I have reduced the length of about 300-400 words. As for English I have resubmitted again the text to a native English language editing service to improve it (please note that yet the first version has been revised by an English language editing service by native English speaker). I have attached the certificates of English editing.  Additionally, the methods for ki-67 assay have been added in the Methods Section.

 My current indications include: 

- line 77, pls remove the second (unnecessary) hyphen; 

Reply: Thank you. I have removed the second hyphen.

- lines 101 and 407: arrange for explaining mTOR with the first use of the abbreviation;

Reply: I have inserted the explanation of mTOR in the first use at line 101: “phosphatidylinositol 3-kinase-Akt-mammalian Target of rapamycin (PI3K/AKT/mTOR)”.

- line 113, pls change ‘the increasing reports that’ to ‘several important reports’;

Reply: I have changed “the increasing reports that” into “several important reports” according to your suggestion.

- line 122, change ‘coexisting any’ to ‘any coexisting’; 

Reply: Thank you. I have changed “coexisting any” into “any coexisting”

- line 128, is: ‘2.2. Measurement’; should be plural;

Reply: I have changed measurement into “measurements”

- line 154, add plural to picogram;

Reply: I have changed “picogram” into “picograms”

- lines 181-183, is: ‘Table 1 and Table 2 report the baseline patients’ clinical and tumor character-istics of endometrial cancer globally and grouped according to cancer subtypes (Type I and II cancer), respectively.’; a better wording would be: ‘Tables 1 and 2 report the patients’ clinical characteristics and studied tumors characteristics, both for the whole group and divided into Type I and II cancer subtypes.’;

Reply: Thank you. I have changed the text as suggested by you as: “Tables 1 and 2 report the patients’ clinical characteristics and studied tumors characteristics, both for the whole group and divided into Type I and II cancer subtypes”

- line 186, of out the sudden ‘Ki-67 expression’ appears; Ki-67 evaluation has not been given in the Methods section; 

Reply: I have added the methodology for Ki-67 evaluation in the Methods Section (see lines 146-157 of the revised version). “Whole section Ki-67 expression in tumor specimen was assessed by immunohistochemistry using the Leica Bond Max (Leica Biosystems, Wetzlar, Germany) with heat-induced epitope regaining. Staining was carried out following the protocol indicated by the International Ki-67 in Breast Cancer Working Group [25]. Slides were stained at room temperature for 1 h with the primary antibody (monoclonal mouse, anti-human Ki-67 antibody; DAKO, Carpinteria, CA), at a dilution of 1:100. Then, the primary antibody was detected using the Refine Detection Kit (Leica Biosystems), that contains a rabbit anti-mouse IgG secondary antibody and anti-rabbit poly-HRP IgG antibody and use 3,3′-diaminobenzidine as a chromogen. The slides were finally counterstained with hematoxylin. The Ki-67 proliferation index was the percentage of positively stained nuclei scored by manual scoring, that calculated the percentage of positively stained nuclei within three high-powered fields (× 40 magnification) randomly identified throughout the tumor, assuring not less than 1000 nuclei were counted [26].”

- Table 3: change ‘(G1 endometrioid) No. 203’ to ‘(G1-2 endometrioid; N = 203) ’, 'No. 102' to 'N = 102', and ‘P value’ to ‘P-value’;

Reply: I have changed the table 3 as indicated by you and for consistency I have also modified the other tables accordingly.

- line 472, „Pointedly’’ is a rarely used word in English language;

Reply: Thank you for your comment. I have eliminated “Pointedly” in the revised version.

- some references have their titles unnecessarily capitalized; some references unnecessarily provide doi numbers. 

Reply: I have eliminated the capitalized titles and the doi numbers, as indicated by you.
